# Androgen Metabolism and Response in Prostate Cancer Anti-Androgen Therapy Resistance

**DOI:** 10.3390/ijms232113521

**Published:** 2022-11-04

**Authors:** Haozhe Zhang, Yi Zhou, Zengzhen Xing, Rajiv Kumar Sah, Junqi Hu, Hailiang Hu

**Affiliations:** 1Department of Biochemistry, School of Medicine, Southern University of Science and Technology, Shenzhen 518055, China; 2Key University Laboratory of Metabolism and Health of Guangdong, Southern University of Science and Technology, Shenzhen 518055, China

**Keywords:** androgen metabolism, androgen receptor, anti-androgen therapy

## Abstract

All aspects of prostate cancer evolution are closely related to androgen levels and the status of the androgen receptor (AR). Almost all treatments target androgen metabolism pathways and AR, from castration-sensitive prostate cancer (CSPC) to castration-resistant prostate cancer (CRPC). Alterations in androgen metabolism and its response are one of the main reasons for prostate cancer drug resistance. In this review, we will introduce androgen metabolism, including how the androgen was synthesized, consumed, and responded to in healthy people and prostate cancer patients, and discuss how these alterations in androgen metabolism contribute to the resistance to anti-androgen therapy.

## 1. Introduction

Prostate cancer (PCa), ranked second in incidence and fifth in cancer-related mortality in men worldwide in 2020 [1], is driven by hormones. Androgens are essential for developing and maintaining normal prostates, benign prostatic hyperplasia (BPH), and PCa [2,3]. Excessive activation of androgen receptors (AR) by androgen is a major factor driving the development and progression of prostate cancer [4].

Androgens are the collective name for a group of 19-carbon steroidal substances, including testosterone (T), dihydrotestosterone (DHT), dehydroepiandrosterone (DHEA), androstenedione, and so on [5]. T and DHT are the main active forms of androgens in the human body [6], and DHT is a more potent AR agonist than testosterone and is the principal androgen bound to AR in the prostate cell [7,8,9]. Androgen/AR signaling plays a critical role in male development from the fetal stage through adolescence into emerging adulthood. Androgen biosynthesis in the human body is an extremely complex process that involves a large system of enzymes. Changes in androgen levels in the body caused by disorders of androgen synthesis can mistakenly activate AR, which has serious consequences for the body [4,10]. For example, the prostate as an androgen-dependent organ can progress into hyperplasia and even cancer by excessive activation of AR [11,12].

Androgen deprivation therapy (ADT) as a mainstay of PCa treatment to reduce the serum testosterone of patients to “castrate” levels (<50 ng/mL), can relieve the disease and improve overall survival when combined with radiotherapy for locally advanced disease, as well as intermediate- and high-risk localized disease [13,14,15]. However, after 2 to 5 years, most patients progress to ADT resistance and castration-resistant prostate cancer (CRPC), characterized by elevated serum prostate-specific antigen (PSA), or the appearance of metastases [8,16]. The occurrence of this resistance is accompanied by changes in the synthesis and metabolism of androgens locally and throughout the lesion. Meantime, the recovery of PSA secretion indicates that AR is reactivated in these cases, enabling cancer cells to respond to castrated levels of androgen [17]. This is mainly caused by the change in AR itself, which is an important inducer of ADT resistance.

CRPC is highly clinically and genetically heterogeneous, which accounts for the emergence of drug resistance and often becomes lethal [18]. A new generation of anti-androgen drugs, represented by abiraterone and enzalutamide was approved by the U.S. Food and Drug Administration (FDA) for CRPC patients a few years ago [19,20,21,22,23]. They work well at first, but patients develop resistance to the drugs within 6–12 months. Multiple mechanisms have been reported to contribute to the anti-androgen resistance in CRPC [24,25]. The androgen metabolism changes and AR variations, which accompany the progression of PCa, are inextricably linked to the development of CRPC drug resistance.

This review will focus on the changes in androgen metabolism, including its synthesis, secretion, and catabolism as well as the variations of AR in response to anti-androgen therapy. We expect that a detailed description of the above processes and changes will fundamentally improve our understanding of the causes of anti-androgen therapy resistance in prostate cancer.

## 2. Androgen Biosynthesis

### 2.1. Enzyme System

Under physiological conditions, androgens are mainly de novo synthesized from cholesterol. Many steroidogenic enzymes are involved in androgen biosynthesis. Two main groups of steroidogenic enzymes are the cytochrome P450 (*CYP450*) enzymes and the hydroxysteroid dehydrogenases (*HSDs*) [26,27].

The *CYP450* enzymes are named because their mixture with carbon monoxide has a specific absorption peak at 450 nm. They are localized in the mitochondria and the endoplasmic reticulum, are hemoglobin-coupled monooxygenases, and function with the participation of coenzyme nicotinamide adenine dinucleotide phosphate (NADP(H)) and molecular oxygen to catalyze many reactions involved in drug metabolism and the synthesis of cholesterol, steroids, and other lipids [28].

*HSDs* catalyze the interconversion between the hydroxysteroid and the ketosteroid, with the participation of NADP(H) or NAD(H). There are four types of *HSDs*: 3β-HSDs, 11β-HSDs, 17β-HSDs, and 20α-HSDs, classified by the carbon number acted upon [29,30,31]. *HSDs* can also be divided into the short-chain dehydrogenase/reductase (SRD) family and the aldol-keto reductase (AKR) family. SDRs act by binding to membrane structures, while AKRs are soluble and play a catalytic role in facilitating the transfer of protons between steroids [32,33].

The androgen synthesis pathway in organisms can be divided into three types: the “classical pathway”, the “alternative pathway”, and the “backdoor pathway”.

### 2.2. Classical Pathway

The first and rate-limiting step in the synthesis of steroid hormones is converting cholesterol into pregnenolone (Figure 1). P450scc is the only known cholesterol side-chain cleavage enzyme in the first reaction and is encoded by the *CYP11A1* gene; P450scc moves to the inner mitochondrial membrane (IMM) and cleaves cholesterol to pregnenolone through a series of electron transfer reactions [34]. Given that the cholesterol concentration in the IMM is usually very low, the efficient transfer of cholesterol from the outer mitochondrial membrane (OMM) to IMM is extremely important for the above reaction to proceed smoothly. Many gonadal and adrenocortical cells express the steroidogenic acute regulatory protein (StAR), which facilitates the transfer of cholesterol from the OMM to the IMM [35].

Next, pregnenolone is first hydroxylated to form 17-hydroxypregnenolone (17-Preg), then cleaved to dehydroepiandrosterone (DHEA) by the steroid 17α-hydroxylase/17,20-lyase (P450c17) which is encoded by the *CYP17A1* gene and located in the endoplasmic reticulum [36]. In parallel to this process, pregnenolone can also be oxidized by 3β-HSDs to progesterone and hydroxylated to 17-hydroxy pregnenolone (17-OHP) by P450c17 and then cleaved to androstenedione (4-AD). This parallel reaction is more often observed in mice. There are two 3β-HSD isozymes, 3β-HSD1 and 3β-HSD2, which are encoded by the *HSD3B1* and *HSD3B2* genes, respectively. 3β-HSD1 is specifically expressed in the placenta and peripheral tissues, including the prostate, while 3β-HSD2 is expressed mainly in the testis, the ovary, and the adrenal [37,38]. In addition, 17-Preg and DHEA are also substrates of 3β-HSDs, and their dehydrogenation by 3β-HSDs results in the formation of 17-OHP and 4-AD [39].

In the human body, DHEA exists primarily as sulfate. DHEA-SO_4_ is the most abundant steroid in male circulation, with about 400–1000 times the concentration of DHEA, and generates free DHEA by the steroid sulfatase [40,41]. DHEA can be catalyzed by 17β-HSD3 or aldol-keto reductase AKR1C3 to produce Δ5-androstenediol (A5diol), then catalyzed by 3β-HSDs to give rise to testosterone. In another way, 4-AD can be directly catalyzed by 17β-HSD3 or AKR1C3 to testosterone. Testosterone can be converted to DHT by Steroid 5-alpha-reductase (S5AR2), an enzyme encoded by the *SRD5A2* gene and mainly expressed in the prostate under normal conditions [42]. In addition to 17β-HSD3, both 17β-HSD2 and 17β-HSD4 play important roles in the classical pathway of androgen synthesis as well and they mainly catalyze the reverse reaction of 17β-HSD3 [43]. These are the classical pathways of androgen synthesis.

### 2.3. Alternative Pathway

In some specific cases, testosterone is not a direct precursor of DHT: DHEA is first converted to 4-AD by 3β-HSDs, then converted to androstenedione by S5AR1, and finally to DHT by 17β-HSD3 or AKR1C3. As this pathway completely bypasses testosterone synthesis, it is known as the “alternative pathway” of androgen synthesis [44,45] (Figure 1). This alternative pathway is dominant in both human CRPC cell lines and human PCa metastases, probably due to the deficiency of 17β-HSD3 and the over-activation of S5AR1, which prefers 4-AD over T, in prostate tumors [46,47].

Interestingly, there are three isoforms of *SRD5A* in human prostate cells, and they have different expression dynamics. *SRD5A2* is predominantly expressed in normal prostate and early-stage PCa, while *SRD5A1* is overwhelmingly predominant in CRPC [44,46]. *SRD5A3* is upregulated in PCa cells following the treatment of *SRD5A* inhibitors, which may result from the activation of AR mutants, but contributes little to androgen synthesis [48].

### 2.4. Backdoor Pathway

Studies of the Macropus eugenii (the smallest species of wallaby) and neonatal genital maturation processes have led to the discovery of the third pathway for androgen synthesis, the “backdoor pathway”, in which 17-OHP is converted to DHT [49,50] (Figure 1). First, 17-OHP is converted to androstanediol (adiol) in the adrenal glands by multistep reactions. Then the adiol is oxidized by 17β-HSD6 to generate DHT in the target tissue. This pathway overcomes the relatively low activity of P450c17 on 17-OHP and therefore does not require DHEA, 4-AD, or T as mediators and is essential for genital development in male Macropus eugenii and occurs in the testes of human fetuses [51,52].

In CRPC, the backdoor pathway is activated because AKR1C3 and 17β-HSD6 play a multi-step catalytic role. There are three isoforms of aldol-keto reductase (AKR1C1, AKR1C2, AKR1C3) [46]: AKR1C1 mainly catalyzes the β-reduction reaction, and the products are mostly biologically inactive, while AKR1C2/3 mainly catalyzes the conversion of DHT to androstanediol, reducing the androgen level [43].

### 2.5. Androgen Synthesis in the Evolution of Prostate Lesions

In healthy men, the de novo synthesis of androgen occurs essentially only in the testes and adrenal glands, while androgens are synthesized from DHEA intake in the prostate through the classical pathway [44]. In an androgen-deficient prostate and BPH, the alternative pathway from DHEA intake becomes active [44] (Table 1).

In castration-sensitive prostate cancer (CSPC), in addition to the classic de novo synthesis of androgen, the alternative pathway turns active. The androgen levels in CSPC cells are elevated relative to normal prostate organs. In response to the ADT treatment, DHT levels are decreased, but the intra-serum T levels of CSPC patients are comparable to those of healthy individuals [47].

In CRPC, the de novo synthesis of androgen through the backdoor pathway also becomes active. After the development of CRPC, the T level in the patient’s lesion was elevated relative to that before ADT treatment, and endocrine therapy was effective in reducing the DHT level in the lesion of CRPC patients [47]. Circulating DHEA-SO_4_ levels are significantly elevated in CRPC patients, for whom the DHEA-SO_4_ synthesis from cholesterol is accelerated in the adrenal glands, and prostate uptake of DHEA is greatly increased [46,48].

## 3. Androgen Secretion and Catabolism

### 3.1. Androgen Secretion

The androgen secretion mainly occurs by the Leydig cells in the testis and is supplemented by the adrenal cortex. The secretion process is regulated by the hypothalamic-pituitary-gonadal axis and the androgens are then transported through the bloodstream to target organs such as the prostate and skin, where androgens enter the cytoplasm in a diffuse manner [53].

Because androgens are hydrophobic, only about 2% of androgens in the blood are in the free state, while most androgens combine reversibly with proteins. In total, 40% of androgens bind to albumins or corticosteroids-binding globulins (CBGs) in a non-specific manner, and 60% of androgens bind specifically to sex hormone-binding globulins (SHBGs) [54]. The protein-bounded androgens temporarily lose their biological activity until unbound androgen is released to the target organs [55]. The prostate tissue is rich in SHBGs, especially androgen-binding proteins (ABPs), which are beneficial to absorb and concentrate androgens [56].

Interestingly, the commensal gut microbiota can promote prostate cancer toward castration resistance by providing an alternative source of androgens [57,58]. Studies have shown that R. gnavus and B. acidifaciens (gut microbiotas) can synthesize alternative sources of androgens that contribute to endocrine resistance in CRPC [59].

### 3.2. Androgen Catabolism

There are two methods of androgen catabolism. One is directly converted to estradiol by aromatase and exerts hormonal effects [60]. Androgens and estrogens have the same precursors. Aromatization of testosterone in peripheral organs is the main source of estrogen in men. The other is converted to hydrophilic metabolites by UDP-glucuronosyltransferase (UGT) enzymes or steroid sulfatase and excreted through urine, bile, and feces. Glucuronidation is the predominant form of androgen catabolism [55].

All androgens are structurally nonpolar lipophilic compounds, and their lipophilicity facilitates their diffusion across biological membranes, making them more accessible to the site of action, but this same property prevents them from being excreted through the kidneys [54]. Therefore, the conversion of nonpolar lipophilic compounds to polar hydrophilic compounds is crucial for the termination of the biological function of the compound, and the metabolism of androgens follows the same principle. The active forms of androgens are all α-configuration products, while α-configuration androgens are metabolized in vivo in a relatively homogeneous manner, mainly through UGT enzymes-mediated glucuronidation [60].

The human UGT enzyme uses UDP-glucuronic acid (UDPGA) as a raw material to covalently attach a glucuronic acid group to a substrate with a suitable functional group, and the mechanism of this reaction is bimolecular nucleophilic substitution (S_N_2) [61]. Many classes of endogenous or exogenous substances, including fatty alcohols, phenols, carboxylic acids, fatty amines, and aromatic amines, can act as receptors for glucuronide groups [56]. For androgens, the glucuronide receptor is the hydroxyl group at its 17- or 3-position [60]. UGT enzymes are a class of transmembrane proteins that, at the cellular level, are mainly localized in the smooth endoplasmic reticulum [62]. Depending on their structure and function, UGT enzymes can be divided into three subfamilies, UGT1A, UGT2A, and UGT2B, all of which are highly expressed in the liver and kidney and perform detoxification functions [62]. For androgen metabolism, UGT2B7, UGT2B15, UGT2B17, and UGT2B28 are the main metabolic enzymes [61].

UGT2B7 is expressed in the liver, kidney, intestine, skin, brain, and breast, but not in the prostate. UGT2B7 has a broader spectrum of catalytic activity for steroids, and it can covalently attach glucuronide groups to the 3-position hydroxyl group of androstanediol or Androsterone [60].

UGT2B15 is expressed in the liver, kidney, skin, breast, uterus, and in prostate organs [62]. In terms of substrate selectivity, UGT2B15 acts on the 17-position hydroxyl group of androstanediol and DHT, while lacking affinity for the 3-position hydroxyl group of steroids [63].

UGT2B17 has high homology with UGT2B15 in amino acid sequence, but they are very different in substrate selectivity. Unlike UGT2B15, UGT2B17 has an affinity for both androgen 17-position and 3-position hydroxyl groups, Androstanediol, Androsterone, and DHT are its substrates. In addition, the spatial expression of UGT2B15 and UGT2B17 in the prostate was also very different, with UGT2B17 being expressed only in prostate basal cells and UGT2B15 being expressed only in prostate luminal cells [63].

UGT2B28, a newly identified androgen metabolizing enzyme, has a similar substrate selectivity to UGT2B17, with an affinity for both the hydroxyl groups at the androgen 17- and 3- positions, it is not involved in the metabolism of DHT but can metabolize T [60] (Figure 2).

## 4. AR Variations in PCa Anti-Androgen Resistance

Androgen acts on prostate normal or cancer cells through its receptor AR. Therefore, the alterations of AR itself and changes in the AR pathway that respond differently to androgen levels will contribute to the anti-androgen resistance in PCa. There are several AR variations, including gene amplification, point mutations, new splicing variants, post-translational modifications, co-activator, and co-repressor modifications, and alternative pathways bypassing AR signaling. These are considered important reasons for anti-androgen resistance [64,65,66,67,68,69,70,71,72,73,74,75,76,77,78,79,80,81,82,83,84,85] (Figure 3).

Anti-androgen therapy, represented by ADT and the endocrine therapeutic drug enzalutamide, can significantly reduce the level of free androgens in patients, thereby decreasing the transcriptional activity of AR. As treatment proceeds, AR itself undergoes multiple changes in response to these changes. They all ultimately result in androgen-independent AR activation, which is the underlying cause of resistance to anti-androgen therapies. The gene amplification and some gain-of-function mutations (such as T878A, F877L W742C, L702H, and so on) can lead AR to be activated in the presence of very little androgens, even in the presence of other hormones even drugs (such as progesterone, flutamide, bicalutamide, and enzalutamide) and maintain transcription activity to promote the tumorigenesis and development of PCa [66,67,68]. Some splicing mutations of AR such as AR-V7 can be activated in the complete absence of androgens and often occur in CPRC patients [69,70]. In addition, there are many post-translational modifications (such as phosphorylation, methylation, and ubiquitination) and AR protein co-activators that can activate AR at low androgen levels [71,72]. Other hormone receptor pathways, including the glucocorticoid receptors (GR) pathway and progesterone receptor (PR) pathway, share similar downstream genes with the androgen receptor pathway, and activation of these bypass pathways is also a possible cause of resistance to anti-androgen therapy [83,84,85].

How to overcome AR alternation during anti-androgen therapy is an enduring topic in prostate cancer resistance research. A deeper understanding of the sources and destinations of androgens may provide new ideas for the field.

## 5. Androgen Receptor Signaling Inhibitors Resistance in PCa

The metabolism of androgens in the human body is the basis for prostate cancer drug development. DHEA is the center of androgen synthesis in the prostate, and the synthesis of DHEA from cholesterol is of great therapeutic importance, especially for PCa patients who have elevated androgen levels and over-activated AR [39]. The National Comprehensive Cancer Network (NCCN)-recommended treatment regimens for PCa, four drugs, abiraterone, enzalutamide, apalutamide, and darolutamide act on the androgen axis, and target the androgen synthesis and metabolism pathways [16]. They are known as androgen receptor signaling inhibitors (ARSI). The reactivation of AR signaling, activation of the bypass pathways, and the recovery of DHEA levels in the body are the main causes of ARSI resistance.

Abiraterone is a selective irreversible inhibitor of the *CYP17A1* enzyme that effectively blocks the synthesis of androgens in the testis, adrenal gland, and PCa [39]. FDA-approved abiraterone acetate with glucocorticoid prednisone acetate for patients with metastatic CRPC (mCRPC) and metastatic high-risk CSPC [16]. Prednisone can reduce the production of corticosterone and 11-deoxycorticosterone and alleviate adverse reactions caused by abiraterone [39]. Available clinical reports show that in combination with prednisone, abiraterone significantly prolongs the overall survival and disease-free survival time of patients with mCRPC [19,20]. However, patients will eventually develop resistance to abiraterone. The causes of resistance to abiraterone are mainly caused by changes in androgen synthesis and metabolic pathways.

Many aspects contribute to abiraterone resistance. Overexpression and mutation of *CYP17A1* and *17β-HSDs* activate the de novo synthesis of androgens and increase the level of DHEA in the tumor [86,87]. Aberrant expression of *3β-HSDs* and *AKR1C3* enhances the “alternative pathway” and “backdoor pathway” of androgen synthesis, respectively, while reducing the metabolism of the active form of androgens by the glucuronidation pathway, resulting in abiraterone resistance [62,87,88,89]. Due to the inhibition of enzyme activity by abiraterone, hormones upstream of P450c17, including adrenocorticotropic hormone (ACTH) and pregnenolone, accumulate, which promotes the enhancement of the “backdoor pathway” of androgen synthesis [90], and pregnenolone itself can also activate mutant AR [91]. The addition of exogenous glucocorticoids to reduce the adverse effects of abiraterone is also thought to activate mutant AR, causing drug resistance [92]. In addition, the emergence of AR truncated variants (such as AR-V7) [93], the activation of the phosphatidylinositol-3-kinase (PI3K)/tyrosine kinase A (AKT)/mammalian target of rapamycin (mTOR) pathway and Erb-B2 receptor tyrosine kinase 2 (ErbB2) pathway also promotes resistance to abiraterone [94].

Enzalutamide is a second-generation androgen receptor antagonist approved by the FDA for patients with both mCRPC and non-metastatic CRPC (nmCRPC) [16]. It inhibits the nuclear translocation of activated AR, preventing its localization to androgen response elements and co-activator recruitment, thereby inducing apoptosis, and inhibiting the proliferation of CRPC cells [95]. Available clinical reports show that enzalutamide significantly prolongs the survival time of prostate cancer patients after chemotherapy [21,22]. However, patients will eventually develop resistance to the drug [24].

Patient resistance to enzalutamide may be influenced by several different mechanisms [24]. Changes in the quantity or structure of AR, including reactivation of AR, production of AR splice variants, and mutations in AR, reduce the overall affinity of enzalutamide for it and are the main cause of enzalutamide resistance. In addition to this, since AR and GR share many common substrates, the over-activation of GR caused by using enzalutamide can also lead to enzalutamide resistance to some extent [96]. Activation of the Wnt pathway and the Warburg effect alters cellular metabolic processes and may also contribute to enzalutamide resistance, as they often end up in aberrant activation of AR signaling independent of androgens [24]. Autophagy inhibition due to the activation of the AMP-activated protein kinase (AMPK) pathway and inhibition of the mTOR pathway also contributes to enzalutamide resistance [97]. Recent studies have also shown that microRNA-mediated gene expression activation can promote neuroendocrine trans-differentiation of CRPC and lead to resistance to enzalutamide [98].

Apalutamide is a novel, second-generation AR antagonist that inhibits the nuclear translocation of AR and inhibits the binding of AR to androgen response-like elements (ARE) in the context of AR expression [99]. Apalutamide is approved for the treatment of nmCRPC and has fewer central nervous system (CNS) adverse effects as compared to enzalutamide [16,99]. Clinical studies have shown that apalutamide has demonstrated an improvement in overall survival (OS) and longer metastasis-free survival in nmCRPC [100,101]. AR mutations and splicing variants are the main causes of apalutamide resistance [102,103]. In addition, increased androgen synthesis within tumors and activation of the PI3K pathway may also cause apalutamide resistance [104,105].

Darolutamide is a novel oral non-steroidal AR inhibitor with a unique chemical structure that binds to AR with high affinity and exhibits strong antagonistic activity, thereby inhibiting receptor function and the growth of prostate cancer cells [106,107,108]. It has been approved for the treatment of nmCRPC and metastatic CSPC (mCSPC) [16,106,107]. Unlike other existing treatments, darolutamide does not cross the blood –brain barrier (BBB), so there are fewer potential drug interactions as well as CNS effects such as epilepsy, falls, and cognitive impairment, limiting the burden that treatment places on patients’ daily lives [109]. There are still few studies on the mechanism of darolutamide resistance, it has been demonstrated that there is a cross-resistance mechanism between darolutamide, apalutamide, enzalutamide, and abiraterone. However, there is evidence that darolutamide had a significative inhibition of the transcriptional activity of AR-mutated variants [109].

## 6. Discussion

The vast majority of prostate cancers are androgen-dependent malignancies, and ADT therapy has been the most important treatment for prostate cancer for more than 70 years [110]. Although early-stage low-grade prostate cancer can be cured with surgery or radiation therapy, most patients have progressed to CSPC by the time they are admitted to the hospital. ADT therapy can promote tumor dormancy, prolong the survival of patients, and improve the quality of life, but it can rarely cure prostate cancer, and most tumors will recur [110,111]. Tumor resistance to ADT is cumulative: with the change of lineage, some tumor cells will eventually lose AR expression and develop into small cell carcinoma; this type of tumor is called neuroendocrine prostate cancer (NEPC), and this type of cancer cells express neuroendocrine markers, mostly induced by treatment, and a small part is primary, and does not respond to ARSI et al. [112]. Developing new treatments for this type of cancer is an urgent problem for basic research. At the same time, tumor classification based on genetic alteration can help guide the clinical precision drug use of prostate cancer. For patients with AR insensitivity, chemotherapy has shown some therapeutic effects [113]. In CRPC patients, about 20% of patients have DNA damage repair (DDR) gene mutations (such as breast cancer type 1 susceptibility protein (*BRCA1*), breast cancer type 1 susceptibility protein (*BRCA2*)) and those patients could be treated by PARP inhibitors (PARPi), which can increase the survival of those patients [114]. Although prostate cancer is a “cold tumor” in the traditional sense of immunotherapy, patients with cyclin-dependent kinase 12 (*CDK12*) and mismatch repair (MMR) gene alterations often benefit from immune checkpoint inhibitor (ICI) therapy [115]. These therapies in combination with ARSI can alleviate cross-resistance. Returning to the status quo of clinical treatment, ADT and ARSI are still the most prevalent therapies, and understanding the details of androgen synthesis and metabolism is critical to addressing the problem of resistance to this type of therapy. We need to be concerned that in most countries, although second-generation hormone therapy drugs are gradually being included in health insurance, the high cost of treatment is still unaffordable for most families [116,117]. The ultimate goal of new therapeutic targets discovery and new drug development should serve patients and their families.

## 7. Conclusions

It is well known that androgen is the ligand of AR to activate AR to take transcriptional function. In the development of PCa, AR always mutated to develop resistance to dealing with various treatments, likely ADT, and antiandrogens. However, in the process to advance prostate cancer, not only AR but also androgen metabolism changed. In healthy men or primary prostate cancer patients, androgen is biosynthesized through the classical or alternative pathway only in the testes and adrenal glands. However, in CRPC patients, the backdoor pathway is activated to synthesize more androgen to meet cancer cell needs. The new generation of hormone therapy drugs could improve the survival time of patients, but it is easy to develop drug resistance. Thus, exploring new methods to treat drug-resistant patients is urgent. A deep understanding of androgen anabolic and metabolic processes in the PCa process, as well as AR changes, helps in the development of new therapies to overcome drug resistance. Multiple metabolic proteins altered in androgens and anabolic changes are also expected to become new drug targets. In CRPC patients, some gene expressions were increased to catabolism of the therapy drugs, such as UTG2B. Developing an inhibitor of those genes might be an opportunity to prolong the drug efficiency for CRPC patients.

## Figures and Tables

**Figure 1 ijms-23-13521-f001:**
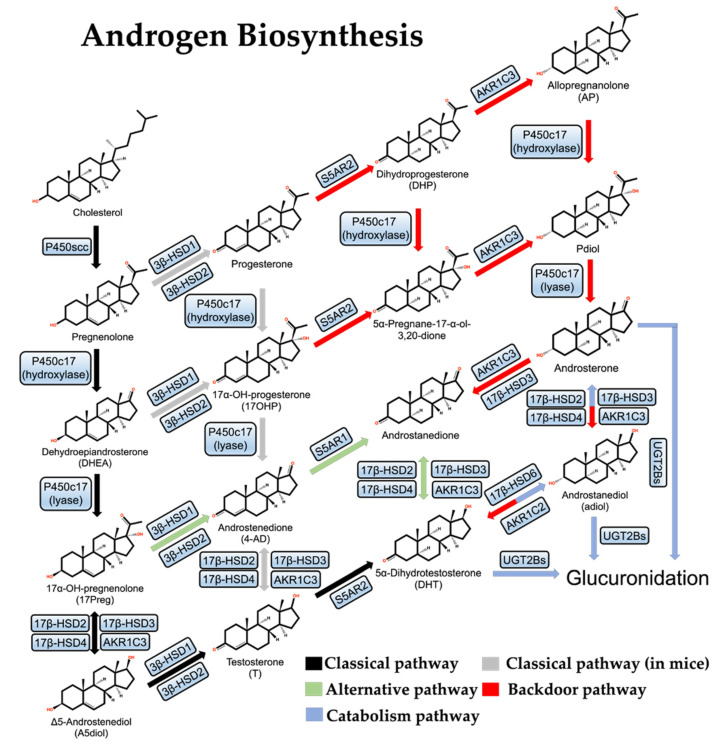
Androgen de novo synthesis and metabolic pathways.

**Figure 2 ijms-23-13521-f002:**
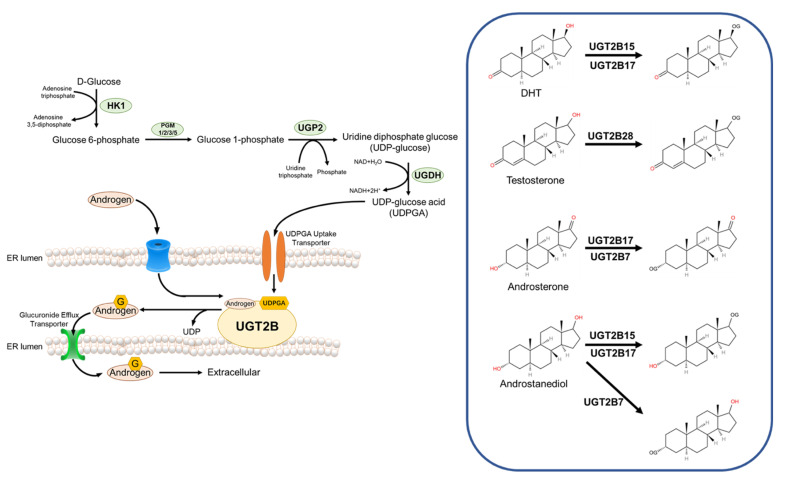
Glucuronidation and androgen catabolism. HK1—Hexokinase 1; PGM—Glucose phosphatase; UGP2—UDP-glucose pyrophosphorylase 2; UGDH—UDP-Glc dehydrogenase.

**Figure 3 ijms-23-13521-f003:**
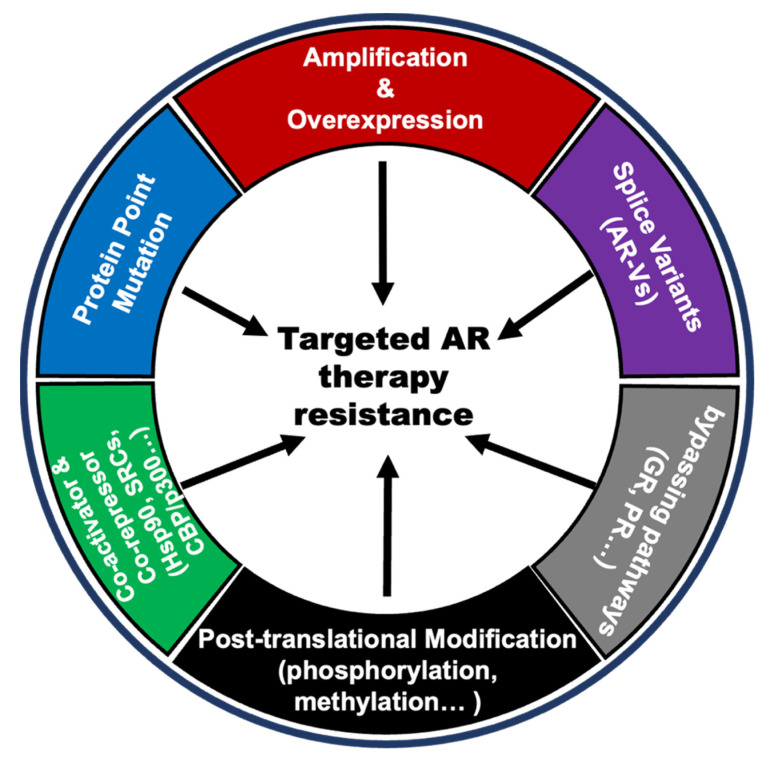
Snapshots of changes in AR and AR pathways in PCa anti-androgen resistance.

**Table 1 ijms-23-13521-t001:** Changes in androgen synthesis pathways in different organs and at different stages of prostate cancer.

	Classical		Alternative		Backdoor	
	De novo Synthesis	DHEA Intake	De novo Synthesis	DHEA Intake	De novo Synthesis	DHEA Intake
Testis	**+**	**+**	**+**	**+**		
Adrenal gland	**+**	**+**	**+**	**+**		
Normal prostate		**+**				
Androgen-deficient prostate		**++**		**+**		
BPH		**++**		**+**		
CSPC	**+**	**++**	**+**	**++**		
CRPC	**++**	**++**	**++**	**++**	**+**	

+ means the pathway begins to appear, ++ means the pathway has been strengthened.

## Data Availability

This study did not report any data.

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
