# Peer review of "Androgen Metabolism and Response in Prostate Cancer Anti-Androgen Therapy Resistance"

_ijms, 2022, doi:10.3390/ijms232113521_

Round 1

Reviewer 1 Report

In this manuscript, the Authors aimed to review the androgen metabolism (synthesis, secretion, and catabolism) in both physiological and pathological conditions, focusing on how alterations of androgen metabolism can contribute to resistance to anti-androgen therapy. The manuscript appears to be composed of two parts. The former is a "biochemical" part (paragraphs 2 and 3), while the second is a "clinical" one (paragraphs 3 and 4).

-The authors provide a detailed and comprehensive review of androgen metabolism in the first part. The figures are great and help the reader to understand the main biochemical processes involved in androgen metabolism.

-The second part is superficial and does not provide thorough information.

o   It is not clear whether the Authors aimed to focus on ARSI mechanisms of resistance in the CRPC or ‘prostate cancer” (PCa) setting. In the first case, the Authors should change the name of paragraphs 4 and 5, changing PCa into CRPC. Furthermore, they must also focus on apalutamide (approved in the nmCRPC setting). In the second case, the Authors have to focus on all the ARSI available for PCa management, including apalutamide and darolutamide (approved in nmCRPC and mHSPC).

o   Although paragraph 4 is named “Abiraterone and Enzalutamide resistance in PCa”, the Authors provide a general description of Abiraterone (Lines 243-256). In contrast, they focus on this drug's resistance mechanisms only from Line 257 to Line 259.

-The Authors did not provide a “Discussion” paragraph. I suggest adding this paragraph in which Authors can speculate about the review results by providing their perspectives. The following references may be helpful for the the "Discussion" and their perspectives: PMID: 31552182; PMID: 33212909; PMID: 34575947; PMID: 35267553; PMID: 35955671.

-In light of the previous point, I suggest removing perspectives from the conclusion paragraph.

Here, I report my minor revisions for the manuscript:

-Line 44: I suggest changing the reference, and providing a citation derived from international guidelines.

-Line 52: I suggest changing references [19] and [20] with some of the clinical trials that led to the approval of these drugs in the CRPC setting (e.g. COU-AA-301; COU-AA-302; AFFIRM; PREVAIL; PROSPER)

-Line 160: I suggest changing from “androgen-dependent prostate cancer” (ADPC) to “castration-sensitive prostate cancer” (CSPC)

-Line 246: I suggest changing this sentence “of the JAMA recommended treatment regimens for PCa”, referring to international guidelines (e.g. NCCN guidelines).

Author Response

Dear Reviewer,

Thanks very much for taking the time to review this manuscript. We have read your comments thoroughly and appreciate all your comments and suggestions. Please find my itemized responses below and my revisions in the re-submitted files.

Line 44 We changed the reference and provided a citation derived from NCCN guidelines.

Line 52 We changed references [19] to [23] into clinical trials including COU-AA-301, COU-AA-302, AFFIRM, PREVAIL, and PROSPER.

Line 161 We changed “androgen-dependent prostate cancer (ADPC)” to “castration-sensitive prostate cancer (CSPC)”

Line 276 We switched Part 4 and Part 5 and changed the title of Part 5 to Androgen receptor signaling inhibitors resistance in PCa

Line 281 We referring to NCCN guidelines.

Line 297-311 We have enriched the content about the abiraterone resistance mechanism.

Line332-352 We added the descriptions of apalutamide and darolutamide.

Line370-399 We have read the literature you suggested and added a " Discussion " section

Thank you again for your comments and suggestions on our manuscript.

Best regards.

Reviewer 2 Report

This review focuses on the changes in androgen metabolism, including its synthesis, secretion, and catabolism as well as the variations of AR in response to anti androgen therapy. A deep understanding of the pathways involved in the development of drug resistance is of the paramount importance to find new treatments for prostate cancer.

The review is well written with illustrations to sum up long paths that would be otherwise difficult to follow when written down. I only have minor comments on this review.

-       At the beginning of the paragraph 2.2 Classical pathways the Authors use a different font.

-       Line 320, paragraph “Conclusions and perspectives”: which can increase their the survival of those patients

Author Response

Dear Reviewer,

Thanks very much for taking the time to review this manuscript. We have read your comments thoroughly and appreciate all your comments and suggestions. We have revised the comments you provided and added some new sections. Please find my itemized responses below and my revisions in the re-submitted files.

Thank you again for your comments and suggestions on our manuscript.

Best regards.

Round 2

Reviewer 1 Report

I congratulate the Authors for their great effort in modifying the manuscript.
Before accepting the paper, just a few minor revisions:

-The "conclusion" paragraph has to be the last paragraph of the review and, thus, should be moved after the "Discussion" paragraph;
-Line 374: "nmCRPC" instead of "n mCRPC";
-Line 417: I suggest adding a reference;
-Line 421: "ARSI et al." instead of "ARSI et all"
-Line 438: I suggest adding a reference.

Author Response

Dear Reviewer,

Thank you very much for your review and approval of this manuscript and your comments. Based on your latest comments, we have made changes.

- We switched the position of the “Discussion” section and the “Conclusion” section.

-Line 337: We changed “n mCRPC” into "nmCRPC";
-Line 360: We added references [111, 112];
-Line 364: We changed "ARSI et all" into "ARSI et al."
-Line 381: We added references [117, 118].

Thank you again for taking the time to review this manuscript and for your suggestions.

Best regards.